

# A molecular phylogenetic appraisal of the acanthostomines *Acanthostomum* and *Timoniella* and their position within Cryptogonimidae (Trematoda: Opisthorchioidea)

Andrés Martínez-Aquino,  Victor M. Vidal-Martínez and
M. Leopoldina Aguirre-Macedo

Departamento de Recursos del Mar, Centro de Investigación y de Estudios Avanzados del Instituto
Politécnico Nacional, Unidad Mérida, Mérida, Yucatán, México

## ABSTRACT

The phylogenetic position of three taxa from two trematode genera, belonging to the subfamily Acanthostominae (Opisthorchioidea: Cryptogonimidae), were analysed using partial 28S ribosomal DNA (Domains 1–2) and internal transcribed spacers (ITS1–5.8S–ITS2). Bayesian inference and Maximum likelihood analyses of combined 28S rDNA and ITS1 + 5.8S + ITS2 sequences indicated the monophyly of the genus *Acanthostomum* (*A.* cf. *americanum* and *A. burminis*) and paraphyly of the Acanthostominae. These phylogenetic relationships were consistent in analyses of 28S alone and concatenated 28S + ITS1 + 5.8S + ITS2 sequences analyses. Based on molecular phylogenetic analyses, the subfamily Acanthostominae is therefore a paraphyletic taxon, in contrast with previous classifications based on morphological data. Phylogenetic patterns of host specificity inferred from adult stages of other cryptogonimid taxa are also well supported. However, analyses using additional genera and species are necessary to support the phylogenetic inferences from this study. Our molecular phylogenetic reconstruction linked two larval stages of *A.* cf. *americanum* cercariae and metacercariae. Here, we present the evolutionary and ecological implications of parasitic infections in freshwater and brackish environments.

## INTRODUCTION

The Cryptogonimidae Ward, 1917, is a speciose family (≥370 species), consisting of 93 genera associated with the intestine or pyloric caeca of marine and freshwater teleosts, reptiles and occasionally amphibians around the world (*Miller & Cribb, 2008a*; *Miller & Cribb, 2013*; *Miller et al., 2009*; *Miller et al., 2010a*; *Miller et al., 2010b*; *Cribb & Gibson, 2017*; *Tkach & Bush, 2010*; *Fernandes et al., 2013*). Since taxonomic identification based on morphological characters is complex (i.e., it is based on combinations of characters), the taxonomic classification of species within Cryptogonimidae (e.g., at the subfamily

Corresponding authors
Andrés Martínez-Aquino,
maandres_@hotmail.com
M. Leopoldina Aguirre-Macedo,
leopoldina.aguirre@cinvestav.mx,
leopoldina2305@gmail.com

level) has been reworked several times (*Miller & Cribb, 2008a*). Taxonomic schemes of subfamilies can also be detected based on ecological factors and host preferences. For example, studies based on phylogenetic approaches infer hierarchical-taxonomic patterns between cryptogonimid species associated with specific marine fish hosts (e.g., *Retrovarium* spp. that are associated with perciform marine fishes), or cryptogonimid genera associated with reptile taxa (e.g., the subfamily Acanthostominae Looss, 1899) (*Brooks, 1980*; *Miller & Cribb, 2007a*; *Miller & Cribb, 2008a*). In particular, the Acanthostominae was inferred based on morphology, phylogeny and biogeographical and host-parasite association patterns (*Brooks, 1980*; *Brooks & Holcman, 1993*). The criteria for the subfamily Acanthostominae, as recognized by *Brooks & Holcman (1993)*, was based on six characters: (1) a terminal oral sucker; (2) a body armed with single row of spines; (3) a preacetabular pit; (4) a genital pore not in preacetabular pit; (5) a seminal vesicle coiled posteriorly; and (6) a sucker-like gonotyl. Based on these criteria, the acanthostomine trematodes include five genera: *Timoniella* Rebecq, 1960; *Proctocaecum* Baught, 1957; *Gymnatrema* Morozov, 1955; *Caimanicola* Freitas and Lent, 1938; and *Acanthostomum* Looss, 1899 (*Brooks, 2004*). Nevertheless, *Miller & Cribb (2008a)* were not convinced by the morphological characteristics that were used to justify subfamily-level divisions in Cryptogonimidae, because several subfamilies were separated by few, and often trivial, characters. *Miller & Cribb (2008a)* also recognized that the phylogenetic analyses of acanthostomines by *Brooks (1980)* could be used to infer intergeneric relationships between cryptogonimids.

To explore the diversity of helminth parasite fauna from aquatic invertebrate and vertebrate hosts in Mexico (*Vidal-Martínez et al., 2001*; *Aguirre-Macedo et al., 2017*), molecular phylogenetic analyses based on nuclear gene fragments (partial 28S ribosomal DNA and the internal transcribed spacers (ITS1–5.8S–ITS2)) were carried out on cryptogonomids from Mexico's Yucatán Peninsula. The analyses were used to answer questions regarding the phylogenetic position of acanthostomines within the family Cryptogonomidae, and possible life-cycle links between cercariae and metacercariae were additionally examined. Based on the results of the molecular phylogenetic analyses, the systematic position of the acanthostomine genera *Acanthosthomum* and *Timoniella* were evaluated, with a brief discussion of the taxonomic implications for the subfamily Acanthostominae, and phylogenetic evidence to support the different intergeneric relationships among Cryptogonimidae is provided.

## MATERIAL AND METHODS
### Collection of hosts and trematode parasites

As part of our ongoing study in the Celestun lagoon (*Sosa-Medina, Vidal-Martínez & Aguirre-Macedo, 2015*), we collected specimens of cryptogonimid metacercariae presumed to be of the subfamily Acanthostominae: *Acanthostomum americanum* (=*Atrophecaecum astorquii*) Pérez-Vigueras, 1956, and *Timoniella* (=*Pelaezia*) *loossi* Pérez-Vigueras, 1956, from the Ria Celestun Biosphere Reserve, Yucatan Peninsula, Mexico (based on *Moravec, 2001*; *Vidal-Martínez et al., 2001*; *Brooks, 2004*; *Miller & Cribb, 2008a*). These metacercariae were collected from the euryhaline fish *Cichlasoma urophthalmus* (Günter, 1862) (Perciformes: Cichlidae) from the Yaxaá water spring (20°53′12.57″N;

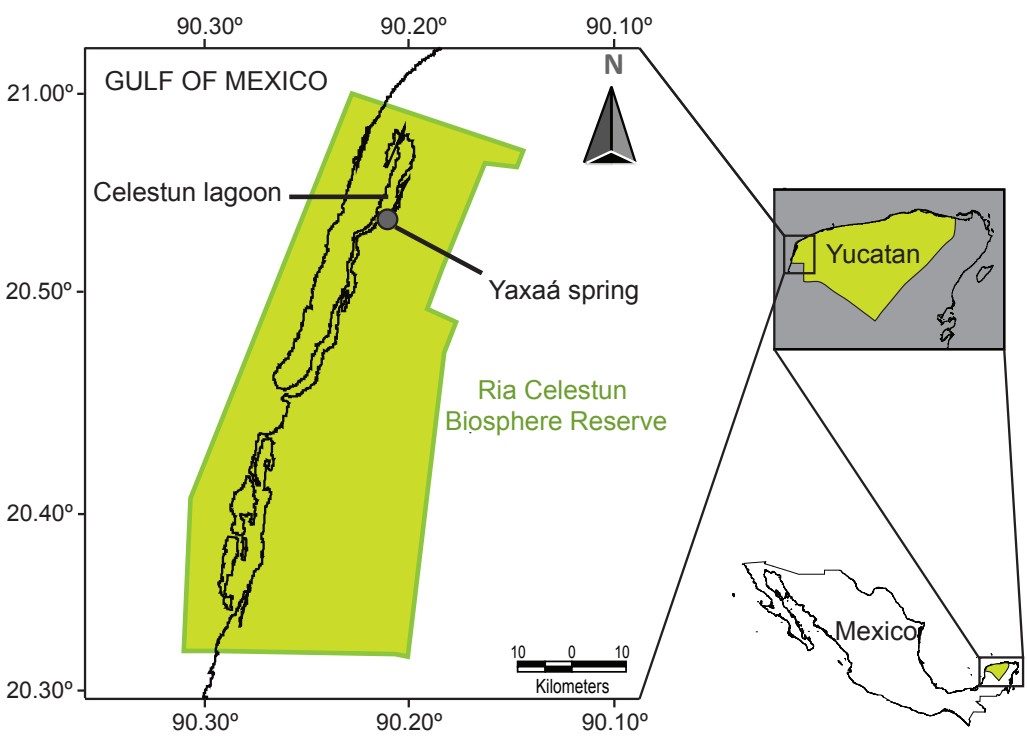

**Figure 1** Map of the study area, Yaxaá spring, Celestun coastal lagoon, Yucatan, Mexico.

90°20′58.86″W), located in the Celestun tropical lagoon (Fig. 1). We also collected cercariae presumed to be of the Cryptogonimidae from the aquatic gastropod *Pyrgophorus coronatus* (Pfeiffer, 1840) (Hybrobiidae) (see *Scholz et al., 2000*), at the same location, to test for possible life-cycle links between the cercariae and metacercariae with molecular data. In March 2016 we collected 223 snails of *P. coronatus* from two localities: Baldiocera Spring (20°54′6.29″N; 90°20′26.46″W) (156 snails) and Yaxaá Spring (67 snails) (the two springs are approximately 1,400 m apart). Snails were collected using strainers, placed separately into glass tubes and maintained in artificial light in the laboratory to stimulate the emergence of cercariae. After 2–3 days, portions of the snails were removed from their shells by dissection under a stereomicroscope. The only representatives of Cyptogonimidae (three cercariae) were collected from a single *P. coronatus* from Yaxaá Spring. For representatives of other families, of the 156 *P. coronatus* examined from Baldiocera Spring, we observed two cercaria of *Ascocotyle* (*Phagicola*) *nana* Ransom, 1920 (Heterophyidae) in each of two individual snails; and one metacercaria of *Crassicutis cichlasomae* Manter, 1936 (Apocreadiidae) from one snail. Both larvae have been previously recorded from *P. coronatus* (*Scholz et al., 2000*). Of the 67 *P. coronatus* examined from Yaxaá Spring, the only cercariae observed belonged to the aforementioned cryptogonimids. We also sampled specimens of other adult cryptogonimids, e.g., *Oligogonotylus mayae* (*Razo-Mendivil, Rosas-Valdez & Pérez-Ponce de León, 2008*), from the cichlid fish *C. urophthalmus*. The protocols for host dissection, examination, collection and preservation, and the

morphological study of parasitic specimens followed *Vidal-Martínez et al. (2001)*. We also collected adult specimens of the apocreadiid species *Crassicutis cichlasomae*, from the same fish host. *Crassicutis cichlasomae* was used as an outgroup taxon for the phylogenetic analyses in this study, based on its previously established sister group relationship of Ophisthorchioidae (*Bray et al., 2009*; *Fraija-Fernández et al., 2015*). Trematodes were identified based on morphological criteria suggested by *Vidal-Martínez et al. (2001)*, *Miller & Cribb (2008a)*, *Razo-Mendivil, Rosas-Valdez & Pérez-Ponce de León (2008)* and *Razo-Mendivil et al. (2010)*. Reliable identification to genus level is possible for both *Timoniella* and *Acanthosthomum* based on metacercariae morphology. Microphotographs of both taxa can be found in Fig. S1. However, identification to species level may be questionable, therefore we hereafter refer to the species as *T.* cf. *loossi* and *A.* cf. *americanum*. Several metacercariae and adult specimens collected for morphological analysis were deposited as voucher specimens [*T.* cf. *loossi* (No. 525), *A.* cf. *americanum* (No. 526), *C. cichlasomae* (No. 527) and *O. mayae* (No. 528)] in the Colección Helmintológica del CINVESTAV (CHCM), Departamento de Recursos del Mar, Centro de Investigación y de Estudios Avanzados del Instituto Politécnico Nacional, Unidad Mérida, Yucatán, México. Acanthostomine cercariae were not deposited because each specimen was required for the molecular study. Comisión Nacional de Acuacultura y Pesca (PPF/DGOPA-070/16) issued the collecting permits.

## DNA extraction, PCR amplification and sequencing

DNA was extracted from individual cercariae, metacercariae and adult trematodes. DNA extraction was performed using the DNAeasy blood and tissue extraction kit (Qiagen, Valencia, CA, USA) following the manufacturer's instructions. For the four trematode taxa, the partial 28S ribosomal gene region was amplified by Polymerase Chain Reaction (PCR) (*Saiki et al., 1988*), using 28sl forward (5′-AAC AGT GCG TGA AAC CGC TC-3′) (*Palumbi, 1996*) and LO reverse (5′-GCT ATC CTG AG(AG) GAA ACT TCG- 3′) (*Tkach, Pawlowski & Mariaux, 2000*). The primers BD1 forward (5′-GTC GTA ACA AGG TTT CCG TA-3′) and BD2 reverse (5′-TAT GCT TAA ATT CAG CGG GT-3′) (*Bowles, Blair & McManus, 1995*) were used for ITS1–5.8S–ITS2 fragment. The reactions were prepared using the Green GoTaq Master Mix (Promega, Madison, WI, USA). This procedure was carried out using an Axygen Maxygen thermocycler. PCR cycling conditions by both molecular markers were as follows: an initial denaturing step of 5 min at 94 °C, followed by 35 cycles of 92 °C for 30 s, 55 °C for 45 s, and 72 °C for 90 s, and a final extension step at 72 °C for 10 min. The PCR products were analysed by electrophoresis in 1% agarose gel using TAE 1× buffer and observed under UV light using the QIAxcel® Advanced System. The purification and sequencing of the PCR products were carried out by Genewiz (South Plainfield, NJ, USA; https://www.genewiz.com/).

## Molecular data and phylogenetic reconstruction

To obtain the consensus sequences of the larvae and adults of *A.* cf. *americanum, T.* cf. *loossi, O. mayae* and *C. cichlasomae,* we assembled and edited the chromatograms of forward and reverse sequences using the Geneious Pro v5.1.7 platform (*Drummond et al., 2010*). To

investigate the monophyly of the taxa included in Cryptogonimidae at the subfamily level, the 28S, ITS1, 5.8S and ITS2 sequences that were generated during this study were aligned with sequences of other cryptogonimids, and their sister groups, heterophyid and opisthorchiid taxa (based on *Thaenkham et al., 2011*; *Thaenkham et al., 2012*), obtained from GenBank (see GenBank accession numbers in Table S1), using an interface available with MAFFT v.7.263 (*Katoh & Standley, 2016*), an "auto" strategy and a gap-opening penalty of 1.53 with Geneious Pro, and a final edition by eye in the same platform. The best partitioning scheme and substitution model for each molecular marker was selected by using the "greedy" search strategy in Partition Finder v.1.1.1 (*Lanfear et al., 2012*; *Lanfear et al., 2014*) and applying the Bayesian Information Criterion (BIC) (*Schwarz, 1978*). The nucleotide substitution model that best fit the 28S data was TVM + I + G (*Posada, 2003*); for ITS1 and ITS2 it was TVMef + G (*Posada, 2003*); and for 5.8S, it was JC + G (*Jukes & Cantor, 1969*). Hypervariable regions of 28S, ITS1 and ITS2 alignments were excluded using the Gblocks Web Server (*Castresana, 2000*; *Talavera & Castresana, 2007*).

The datasets were analysed by Bayesian inference (BI) and Maximum likelihood analyses (ML) using the CIPRES Science Gateway v. 3.3 (*Miller, Pfeiffer & Schwartz, 2010*). ML analyses were conducted in RaxML v. 8 (*Stamatakis, 2014*) using the GTRCAT approximation as a model of nucleotide substitution (*Yang, 1994*; *Yang, 1996*; *Stamatakis, 2006*). BI analyses were carried out with MrBayes v. 3.2.1 (*Ronquist et al., 2012*). The Bayesian phylogenetic trees were reconstructed for each gene separately using two parallel analyses of Metropolis-Coupled Markov Chain Monte Carlo (MCMC) for $20 \times 10^6$ generations each. Topologies were sampled every 1,000 generations and the average standard deviation of split frequencies was observed until it reached <0.01, as suggested by *Ronquist et al. (2012)*. A majority consensus tree with branch lengths was reconstructed for the two runs after discarding the first 5,000 sampled trees. For both ML and BI analyses, model parameters were independently optimized for each partition. Node support was evaluated by non-parametric bootstrapping (*Felsenstein, 1985*) with 1,000 replicates performed with RAxML (ML) and BI by Posterior probabilities (PP), where bootstrap values ≥75% and PP ≥ 0.95, were considered strongly supported.

## RESULTS

### DNA sequences and dataset analyses

In total, 36 bi-directional partial 28S (domains 1 and 2) and ITS1-5.8S-ITS2 sequences were obtained from three individual cercariae and three individual metacercariae from *A.* cf. *americanum*, as well as three individual metacercariae from *T.* cf. *loossi*, *O. mayae* (one adult specimen), and *C. cichlasomae* (one adult specimen, outgroup) (Table 1). The partial 28S rDNA sequence fragment consisted of 881 base-pairs (bp) for the cercariae and metacercariae of *A.* cf. *americanum*; 880 bp in *T.* cf. *loossi*, 871 bp in *O. mayae*, and 870 bp in *C. cichlasomae*. The 28S sequences of cercariae and metacercariae of *A.* cf. *americanum* from *P. coronatus* were identical, while the sequences of *T.* cf. *loossi* showed a divergence of 0.03%. Nucleotide sequence variation in the 28S alignment from cryptogonimids (excluding the outgroup taxon) from 28S included 722 conserved sites, 537 variable sites,
**Table 1  GenBank accession numbers for cryptogonimid species sequences newly generated for this study.** Codes used for each cryptogonimid sequenced are as shown in the terminal taxa names of Fig. 1 and Figs. S2–S4.

| Name | Code | Life cycle stage | GenBank accession | |
|---|---|---|---|---|
| | | | 28S | ITS1-5.8S-ITS2 |
| *Timoniella* cf. *loossi* | 1 | Metacercarie | MG383502 | MG383515 |
| *Timoniella* cf. *loossi* | 2 | Metacercarie | MG383503 | MG383516 |
| *Timoniella* cf. *loossi* | 3 | Metacercarie | MG383504 | MG383517 |
| *Timoniella* cf. *loossi* | 4 | Metacercarie | MG383505 | MG383518 |
| *Timoniella* cf. *loossi* | 5 | Metacercarie | MG383506 | MG383519 |
| *Acanthostomum* cf. *americanum* | 1c | Cercarie | MG383496 | MG383509 |
| *Acanthostomum* cf. *americanum* | 2c | Cercarie | MG383497 | MG383510 |
| *Acanthostomum* cf. *americanum* | 3c | Cercarie | MG383498 | MG383511 |
| *Acanthostomum* cf. *americanum* | 1m | Metacercarie | MG383499 | MG383512 |
| *Acanthostomum* cf. *americanum* | 2m | Metacercarie | MG383500 | MG383513 |
| *Acanthostomum* cf. *americanum* | 3m | Metacercarie | MG383501 | MG383514 |
| *Oligogonotylus mayae* | | Adult | MG383507 | MG383520 |
| *Crassicutis cichlasomae* | | Adult | MG383508 | MG383521 |

403 parsimony-informative sites, and 134 singleton sites. The sequence fragments for the ITS1 nuclear marker were between 709 and 781 bp in length for *A.* cf. *americanum*; and were 805 bp in *T.* cf. *loossi*, 613 bp in *O. mayae*, and 424 bp in *C. cichlasomae*. The 5.8S nuclear marker was composed of 160 bp in *A.* cf. *americanum*, *T.* cf. *loossi*, *O. mayae* and *C. cichlasomae*. The length of the ITS2 nuclear marker ranged from 259 bp to 277 bp in *A.* cf. *americanum* and from 268 bp to 277 bp in *T.* cf. *loossi*; 260 bp in *O. mayae*, and 295 bp in *C. cichlasomae*. The ITS1 and ITS2 sequences of *A.* cf. *americanum* displayed 4% and 0.7% divergence, respectively, and those from *T.* cf. *loossi* displayed 0.9% divergence and 100% pairwise identity; the 5.8S sequences were identical. Nucleotide sequence variation (excluding the outgroup taxa) for ITS1, 5.8S and ITS2 were 62/69/50 conserved, 406/92/212 variable, 341/36/184 parsimony-informative, and 65/56/28 singleton sites, respectively.

## Phylogenetic reconstructions

We inferred the phylogenetic relationships of Cryptogonimidae, based on the BI and ML analyses, from the following two datasets. The partial 28S gene dataset contained 92 terminals belonging to 81 species, and the combined dataset (28S + ITS1 + 5.8S + ITS2) contained 294 sequences belonging to 81 taxa concatenated (all sequences available from GenBank, see Table S1). The phylogenetic trees constructed from the 28S and the concatenated datasets (28S + ITS1 + 5.8S + ITS2), based on BI and ML analyses, were broadly congruent. For example, all clades with high nodal support values (PP $\geq$ 0.95 and bootstrap $\geq$ 75%) and analysed with the concatenated and 28S datasets were recovered with both BI and ML (Fig. 2; Figs. S2–S4). Only three clades were recovered with high nodal support values (PP $\geq$ 0.95) using BI but not ML (i.e., (*Gynichthys diadikidnus*, *Neoparacryptogonimus ovatus*); (*Metagonimus takahashii*, *M. yokogawai*); and (*Haplorchis*

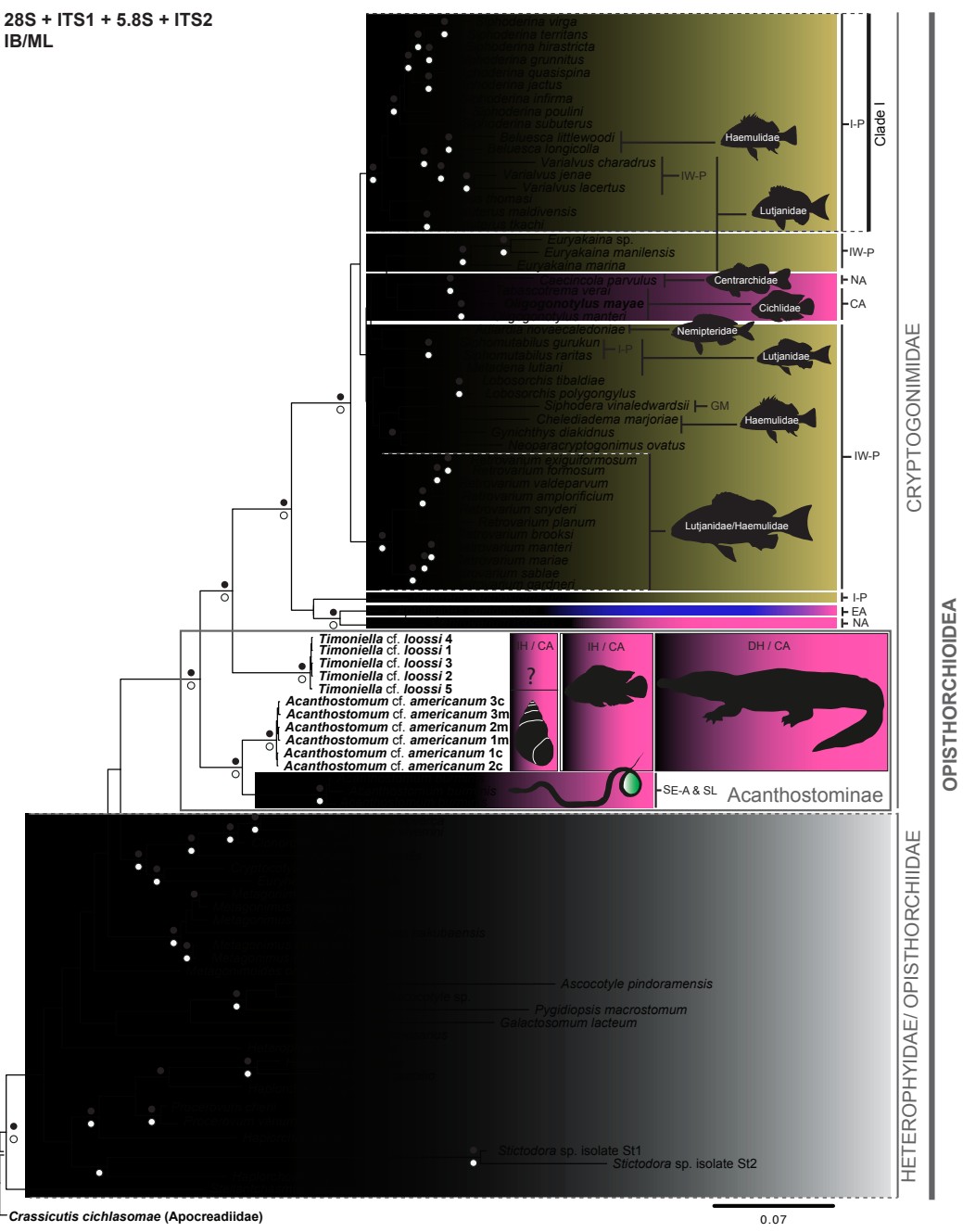

**Figure 2** **Phylogenetic tree obtained from Bayesian inference analysis of the concatenated data (28S + ITS1 + 5.8S + ITS2) of species of the Cryptogonimidae.** The scale bar represents the number of nucleotide substitutions per site. Codes following taxon names are cross-referenced in Table 1 and Table S1. Filled black circles above and white circles below the branches represent Bayesian posterior probability ≥ 0.95 and Maximum likelihood bootstrap support values ≥ 75%, respectively. Diffused green, Freshwater environment; Diffused green-yellow, Brackish environment; Diffused blue, Marine environment; IH, Intermediate host; DH, Definitive host; ?, Intermediate host unknown. (continued on next page...)

**Figure 2 (…continued)**
I-P, Indo-Pacific; IW-P, Indo-west Pacific; CA, Central America; GM, Gulf of Mexico; NA, North America; EA, Eastern Atlantic; Se-A & SL, South-eastern Asia and Sri Lanka. The black snail outline corresponds to *Pyrgophorus coronatus*. The black fish outline corresponds to *Cichlasoma urophthalmus*. The black crocodile outline corresponds to *Crocodylus moreletii*. The black fishes outline on the remaining Cryptogonomidae refer to host specificity at family (ies) recording to species, species groups or genus (black line) of cryptogonomids. The animals' silhouettes were modified from *Ditrich et al. (1997)* (snail); *Gray (1830)* (snake); *Nelson (2006)* (fishes), and *Sánchez Herrera et al. (2011)* (crocodile). The cryptogonomid taxa without black fish outline are not specific to one host. See text for more details.

*yokogawai* (*Haplorchis popelkae, Haplorchis pumilio*))), while only one clade received high nodal support value (bootstrap ≥ 75%) with ML and not BI (i.e., (*Haplorchoides* sp. (*Stictodora* sp. isolate St1, *Stictodora* sp. isolate St2))) (Fig. 2). Conversely, only one difference was observed between the topology of the phylogenetic trees obtained from the 28S and concatenated datasets with BI and ML. Namely, the phylogenetic tree obtained from the ML analysis of the 28S sequence dataset contained a polyphyletic group (without nodal support value), i.e., *Siphodera vinaledwardsii*, *Gynichthys diakidnus*, *Chelediadema marjoriae*, *Caecincola parvulus*, and *Tabascotrema verai* (Fig. S3). In all trees, acanthostomines were paraphyletic, with high nodal support values (PP ≥ 0.95 and bootstrap ≥ 75%). Based on all trees, the family Cryptogonimidae appears to have arisen from a paraphyletic Heterophyidae/Opisthorchiidae group. As well, all trees clearly showed that the generated sequences in this study of *T.* cf. *loossi* and *A.* cf. *americanum* form a monophyletic group with high nodal support values (PP ≥ 0.95 and bootstrap ≥ 75%), respectively. These acanthostomine genera are sister to the remaining cryptogonomids. Furthermore, the genus *Acanthostomum* is a monophyletic group with high nodal support values (PP ≥ 0.95 and bootstrap ≥ 75%). Lastly, the 28S and ITS1-5.8S-ITS2 fragment sequences of acanthostomine metacercaria from *C. urophthalmus* were identical to those of cercarie from *P. coronatus*, and therefore both trematode stages correspond to the same taxa, *A.* cf. *americanum*.

The phylogenetic relationships among Cryptogonimidae at the generic level had high support (PP ≥ 0.95) and the genera *Siphoderina*, *Belusca*, *Varialvus*, *Caulanus* and *Latuterus* form a monophyletic group (Clade I) (Fig. 2), distributed in the Indo-Pacific region (I-P). Of these, the genera *Belusca* and *Varialvus*, *Caulanus*, and *Latuterus* have been found parasitizing the marine fish families Haemulidae and Lutjanidae. Furthermore, *Retrovarium* spp. was found parasitizing Lutjanidae and Haemulidae from the Indo-West Pacific (IW-P) (Fig. 2).

## DISCUSSION

The phylogenetic trees obtained from BI and ML analyses, inferred from the 28S and concatenated dataset, identified the phylogenetic position of the acanthostomines *A.* cf. *americanum* and *T.* cf. *loossi,* and illustrate different intergeneric relationships among cryptogonomids. Phylogenetic analyses show that the Heterophyidae and Opisthorchiidae are paraphyletic as previously reported (*Thaenkham et al., 2011*; *Thaenkham et al., 2012*; *Fraija-Fernández et al., 2015*; *Stoyanov et al., 2015*; *Borges et al.,*

*2016*), and that the family Cryptogonimidae appears to have arisen from the paraphyletic Heterophyidae/Opistorchiidae. This phylogenetic inference is based on a dataset of 51 taxa of Cryptogonimidae that included 24 genera. At present, the family Cryptogonimidae includes 93 genera (*Cribb & Gibson, 2017*), and we analysed almost 40% (38.75%) of recorded genera of Cryptogonimidae. Therefore, the phylogenetic inference of Cryptogonimidae has an appropriate taxonomical representation, but it is still necessary to complete this work with more sampling and sequencing of the remaining non-investigated genera.

Based on the phylogenetic position of *A.* cf. *americanus*, *A. burminis* (which formed a single clade) and *T.* cf. *loossi* (independent lineage), we find that the subfamily Acanthostominae is paraphyletic. Therefore, the monophyly proposed for the subfamily Acanthostominae based on morphological analyses (i.e., *Brooks, 1980*; *Brooks, 2004*; *Brooks & Caira, 1982*; *Brooks & Holcman, 1993*) does not appear to be valid. These data support the proposed invalidity of the subfamily-level division of Acanthostominae into Cryptogonimidae, as previously suggested by *Miller & Cribb (2008a)*. Therefore, it is necessary to include more acanthostomine taxa (i.e., *Proctocaecum*, *Gymnatrema*, *Caimanicola*) in future studies to determine their phylogenetic positions and test their monophyly.

Based on the phylogenetic positions of *Acanthostomum* spp. and *T. loossi* in this study, we postulate a probable host-specificity pattern at a supra-specific level. The adult trematodes *A. burminis*, *A. americanum* and *T. loossi* are associated with freshwater diapsid sauropsids, i.e., *Xenochrophis piscator* (Schneider, 1799) (snake) (Reptilia: Colubridae) and *Crocodylus moreletii* Duméril & Bibron, 1851 (crocodile) (Reptilia: Crocodylidae) (*Moravec, 2001*; *Jayawardena et al., 2013*; *Sosa-Medina, Vidal-Martínez & Aguirre-Macedo, 2015*). The molecular evidence that links the two larval stages of *A. americanum* to the freshwater environment (from their intermediate hosts: snails and fish) and their later development as adults in freshwater crocodiles, may reflect an ecological preference to a freshwater environments. More specifically, the first larval stage (i.e., cercaria) of *A.* cf. *americanum* is restricted to freshwater environments due to the intermediate host snail's intolerance to brackish water (*Scholz et al., 2000*). The trematode's intermediate and definitive vertebrate hosts (*Cichlasoma urophthalmus* and *Crocodylus moreletii*) are both tolerant to brackish water and can move between the two aquatic environments (*Platt, Sigler & Rainwater, 2010*; *Miller et al., 2009*); however, the freshwater environment is essential to completing the trematode's life cycle. This assertion is supported by taxonomic records of metacercariae of *A.* cf. *americanum* being from freshwater fishes of the families Characidae, Cichlidae, Clupeidae and Poeciliidae (*Salgado-Maldonado, 2006*; *Sosa-Medina, Vidal-Martínez & Aguirre-Macedo, 2015*).

Our phylogenetic trees indicated that the Acanthostominae were sister to the remaining marine cryptogonimids (supporting the sister-group relation found by *Stoyanov et al., 2015*) (Fig. 2). If the acanthostomine taxa are truly sister to the remaining Cryptogonomidae, there would be a strong argument for the hypothesis that the cryptogonimids originated in a freshwater environment and later diversified and colonized brackish and marine environments. The transition from the freshwater environment to
the brackish and marine environments is an evolutionary process also inferred for other platyhelminth groups (e.g., *Torchin, Lafferty & Kuris, 2002*; *Boeger, Kritsky & Pie, 2003*; *Van Steenkiste et al., 2013*). Future studies may test this hypothesis regarding the colonization from freshwater to marine environments (e.g., *Waters & Wallis, 2001*; *Grosholz, 2002*; *Lee & Gelembiuk, 2008*). The identification of the link between the cercariae and metacercariae of *A.* cf. *americanum* may represent a step in the understanding of the evolutionary strategies employed within different aquatic environments and the potential repercussions on food webs (e.g., *Shoop, 1988*; *Dobson, Lafferty & Kuris, 2006*; *Poulin, 2006*).

It is noteworthy that the hydrobiid snail *P. coronatus* is highly susceptible to trematode infection, as it has been reported to harbour 12 trematode species, i.e., *Genarchella astyanactis* Watson 1976; *Echinochasmus leopoldinae* Scholz et al. 1996; *Echinochasmus macrocaudatus* Ditrich et al. 1996; *Saccocoelioides* cf. *sogandaresi* Lumsden 1963; *Crassicutis cichlasomae* Manter 1936; Homalometridae gen. sp.; *Oligogonotylus manteri* Watson 1976; *A.* (*Phagicola*) *nana* Ransom 1920; *Ascocotyle* (*Ascocotyle*) sp.; Xiphidiocercaria type 1, Xiphidiocercaria type 2 and Xiphidiocercaria type 3 (*Scholz et al., 2000*). The record of *A.* cf. *americanum* in *P. coronatus* is a new cercaria record for this snail. However, unfortunately, we did not collect sufficient cercariae of *A.* cf. *americanum* to describe their morphology.

Our analyses recovered a monophyletic group (Clade I) that includes *Belusca*, *Caulanus*, *Latuterus*, *Siphoderina* and *Varialvus* distributed in the Indo-Pacific (*Miller & Cribb, 2007a*; *Miller & Cribb, 2008b*; *Miller et al., 2010b*) (Fig. 2). Based on the diversity of genera in this clade, possible taxonomic implications include the erection of a new taxonomic hierarchy at the subfamily level. Future studies based on morphological evidence may support or reject this taxonomic inference. Presently, more than 50 cryptogonimid taxa have been recorded from fishes belonging to the Lutjanidae and Haemulidae of the IW-P (*Miller & Cribb, 2007b*; *Cribb et al., 2016*), reflected in the phylogenetic topology revealed in this study; e.g., the genera *Beluesca*, *Varialvus*, *Caulanus*, *Latuterus*, *Siphomutabilus*, *Metadena*, *Chelediadema*, and *Gynichthys* (Fig. 2) (*Miller & Cribb, 2007c*; *Miller & Cribb, 2009*; *Miller & Cribb, 2013*; *Miller et al., 2010a*; *Miller et al., 2010b*; *Miller, Bray & Cribb, 2011*; *Overstreet, Cook & Heard, 2009*). Furthermore, *Adlardia novaecaledoniae* has been found in Nemipteridae from the Indo-West Pacific (*Miller et al., 2009*). On the other hand, previous records of *Euryakaina* spp. and *Retrovarium* spp. have been found in the families Lutjanidae and Haemulidae families from the Indo-West Pacific, and were attributed by the authors to a host specificity pattern at the supra-specific level (*Miller & Cribb, 2007b*; *Miller et al., 2010a*; *Miller, Bray & Cribb, 2011*). Similarly, cases of monophyletic groups from this study originating from specific families could indicate cases of host specificity (probably resulting from co-divergence *Page, 2003*; *Martínez-Aquino, 2016*), although we cannot rule out the possibility of these cases are an artefact of sampling bias. Future taxonomical studies of cyryptogonomid trematodes from marine fishes from other parts of the world will shed more light on host-specificity patterns (e.g., *Barger, 2010*; *Montoya-Mendoza et al., 2014*).

Additionally, several non-acanthostomine cryptogonimid clades associated with the freshwater environment are specialist parasites of particular families of freshwater fishes from North and Central America; e.g., *Caecincola parvulus* is associated with Centrarchidae

from North America (NA), and *Tabascotrema verai*, *O. mayae* and *O. manteri* are associated with Cichlidae from Central America (CA) (*Choudhury et al., 2016*). Even though these groups did not have valid nodal support in this study (Fig. 2), it is important to mention three points. First, the freshwater cryptogonimids appear to have arisen from among the marine taxa. Second, *C. parvulus* and *Oligogonotylus* spp. occur in freshwater fishes as both adults and metacercariae (*Stoyanov et al., 2015*; *Choudhury et al., 2016*). Third, considering that centrarchids and cichlids are both members of Percomorpha and have marine affinities, *Choudhury et al. (2016)* suggested that a close relationship exists between Middle-American cryptogonimids of cichlids and cryptogonimids of North American centrarchids. The phylogenetic relationship we found between cryptogonimids of cichlids and centrarchids supports this hypothesis. However, recent records of *C. parvulus* from other freshwater fish families must also be considered before final conclusions are made (*McAllister et al., 2015*; *McAllister et al., 2016*).

Studies of cryptogonimids (and trematodes in general) are negatively impacted by the lack of taxonomical records of helminth parasites of freshwater and marine fishes of different regions (*Scholz & Choudhury, 2014*; *Cribb et al., 2016*; *Vidal-Martínez, Torres-Irineo & Aguirre-Macedo, 2016*), as well as the lack of knowledge concerning intermediate and definitive host life cycles (*Cribb & Bray, 2011*; *Blasco-Acosta & Poulin, 2017*). This has led to a reduction in postulated evolutionary hypotheses on the diversification patterns of parasites. However, the development of phylogenetic hypotheses as presented here can provide a modern framework in parasite evolutionary ecology (e.g., *Littlewood, 2011*; *Gómez & Nichols, 2013*; *Poulin, Blasco-Costa & Randhawa, 2016*).

## ACKNOWLEDGEMENTS

Thanks are due to staff of the laboratory of Patología Acuática: Clara Vivas Rodríguez, Gregory Arjona-Torres, Ana L. May-Tec, Francisco Puc Itzá, Nadia Herrera Castillo, Jhonny G. García-Teh, Arturo Centeno-Chale, Germán López-Guerra, Daniel Aguirre-Ayala and Efraín Sarabia from CINVESTAV-IPN, Unidad Mérida, México. We are grateful to Abril Gamboa-Muñoz and José García Maldonado for their technical assistance in the molecular lab. We also thank Dr. Fadia Sara Ceccarelli who reviewed the first draft of this manuscript and made very useful suggestions of the phylogenetic analyses that we present in this contribution. The manuscript greatly benefited from comments of Dr. Terry Miller and two anonymous referees.

### Funding

Financial support of Andrés Martínez-Aquino at the laboratory of Aquatic Pathology at CINVESTAV–IPN came from Grant No. 201441 financed by the Sectorial Hidrocarbon Fund CONACYT-SENER to the CIGoM Consortium. The funders had no role in study design, data collection and analysis, decision to publish, or preparation of the manuscript.

## Grant Disclosures

The following grant information was disclosed by the authors:
Sectorial Hidrocarbon Fund CONACYT-SENER: 201441.

## Competing Interests

The authors declare there are no competing interests.

## Author Contributions

- Andrés Martínez-Aquino conceived and designed the experiments, performed the experiments, analyzed the data, wrote the paper, prepared figures and/or tables, reviewed drafts of the paper.
- Victor M. Vidal-Martínez contributed reagents/materials/analysis tools, wrote the paper, reviewed drafts of the paper.
- M. Leopoldina Aguirre-Macedo conceived and designed the experiments, contributed reagents/materials/analysis tools, wrote the paper, reviewed drafts of the paper.

## Field Study Permissions

The following information was supplied relating to field study approvals (i.e., approving body and any reference numbers):

Comisión Nacional de Acuacultura y Pesca (PPF/DGOPA-070/16) issued the collecting permits.

## Data Availability

The sequences are provided as a Supplemental File.

## Supplemental Information

Supplemental information for this article can be found online at http://dx.doi.org/10.7717/peerj.4158#supplemental-information.

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
