# Peer review of "A molecular phylogenetic appraisal of the acanthostomines Acanthostomum and Timoniella and their position within Cryptogonimidae (Trematoda: Opisthorchioidea)"

_PeerJ, doi:10.7717/peerj.4158_

## Round 0.1 · original submission · Minor Revisions

I have heard back from two reviewers, both of whom feel your work will be a good addition to the literature. They have both offered some comments to help improve your work, and hence my decision is "minor revisions" are needed. I look forward to seeing your revised version.

Reviewer 1 ·

Basic reporting

no comment

Experimental design

no comment

Validity of the findings

no comment

Additional comments

The MS titled “A molecular phylogenetic appraisal of the acanthostomines Acanthostomum and Timoniella and their position within Cryptogonimidae (Trematoda: Opisthorchioidea)” is a well-written, insightful examination of the position of two acanthostomine genera. I found the MS to be clear and well referenced, the data is robust and the images attractive and representative of the data. I have a few comments regarding the MS, and pending consideration of these comments I recommend this article be published in PeerJ

Major points
1) The authors do not discuss the position of the acanthostomine genera in the phylogenetic results section, it is almost entirely discussing the associations of those sequences from GenBank. Seeing as this is the basis of the paper I think it needs to be covered in the results section.
2) On line 193 the author mention “Only three high nodal support values (PP ≥ 0.95) from three clades”, yet later in the pargraph they mention that several other clades are support values (PP ≥ 0.95) (line 204 and 208). This needs to be clarified
3) Lines 207-217 and 279-298 discuss clading of species based on host associations or geographical localities. I strongly recommend these be reduced and extensively edited, as these associations are not really evident in the data. No clades can be found by geography (e.g. there are species from the IWP/IP (which is almost the same thing) across the marine crypto clade that do not form any really grouping) or host (e.g. there are species from Lutjanids or Haemulids across the marine crypto clade that do not form any really grouping). It is highly likely that these clades actually represent the selective sampling or Miller et al, from which most of these sequences were generated, in that the studies focused on certain host groups (reef fishes of a few families) in certain locations (GBR/French Polynesia).

Minor points
Line 15, et al: I think the authors need to make sure they say (at least a few times) that It is partial 28S rDNA, as only 880ish bases (Domains 1-2) of the region were sequenced

Line 19: “paraphyly of the genera Acanthostomum and Timoniella acanthostomines” makes it sound the like genera are paraphyletic. I would change to “paraphyly of the Acanthostominae”

Line 20: change “by itself” to “alone”

Line 46: change to “1) a terminal oral sucker; 2) a body armed with a single row of spines; 3) a preacetabular pit; 4) a genital pore that is not in preacetabular pit; 5) a seminal vesicle that is coiled posteriorly; and 6) a sucker-like gonotyl.”

Line 53: comma after Cryptogonimidae; change to “by few, and often trivial, characters”

Line 60: remove comma after “astorquii)”

Lines 57-74: Some of this paragraph should be moved to the Methods section, it is not an introduction to the subject

Line 84: “hydrobiid snails of P. coronatus” to “snails of P. coronatus (Hydrobiidae)”

Line 86: “The snails” should be “Snails”

Line 90: “As for” should be “For”

Line 94: “were” should be “have been”

Line 100: change “trematode species from the same fish host, Crassicutis cichlasomae” to “trematode species, Crassicutis cichlasomae, from the same fish host”

Lie 105: “The identification to genus level for both Timoniella and Acanthosthomum is certain based on metacercariae morphology” should be changed to “Reliable identification to genus level is possible for both Timoniella and Acanthosthomum based on metacercariae morphology”

Line 107: “questioned” should be “questionable”

Line 108: “metacercarian” should be “metacercarial”

Line 124: insert “fragment” after “ITS1–5.8S–ITS2”

Line 148: “with Bayesian” should be “by Bayesian”; add “analyses” after “(ML)”

Line 149: change “The ML was” to “ML analyses were”

Line 151: change “The BI was” to “BI analyses were”

Line 169: delete “those of”

Line 203: change “acanthostomines form a paraphyletic group” to “acanthostomines are paraphyletic”
Line 204: delete “with Acanthostomum and Timoniella not clustering together”. This is superfluous

Line 224: change “…2016). The…”to “…2016), and that the…”

Line 227: change “It’s indicate that” to “This indicates that”

Line 229: “an appropriate”

Line 232: “separate” is implied. Delete.

Line 251: I could put the genus names in full here, seeing as the are both C.

Line 254: Maybe the authors mean “from freshwater regions”, they just noted that C. urophthalmuscan tolerate brackish water so all records can’t be from “only freshwater fishes”

Line 257: change “Acanthostominae was a freshwater group that was sister to the remaining” to “Acanthostominae were sister to the remaining”. As the two genera represented independent lineages, you cannot say “freshwater group”

Line 260: Cryptogonimidae is misspelled

Line 271: add comma after “infection”; change “e.g.” to “i.e.” as you list all 12

Line 273: is the “?” meant to be in this species name

Line 305: “arisen” rather than “arise”

Line 311: “suggested” rather than “suggest”

Line 314: delete “would”

Figure 2: What do the larger black circle inside the nodes indicate?

Table 1: the species name in the second line of table needs to be italicized

Other trees: why are the species not “cf.” in these trees

·

Basic reporting

No comment

Experimental design

No comment

Validity of the findings

No comment

Additional comments

This is a well-written and presented paper describing the relationships of species of Acanthostomum and Timoniella to other previously reported cryptogonimid genera for which comparative molecular data is available.

The methods used for molecular characterisation of the cercariae, metacercariae and adults reported here are appropriate and sufficiently detailed.

Overall, this is a solid paper and I have no major issues with the work presented.

However, I do have some minor comments and edits that the authors should address prior to publication. I have included these as annotations and comments on the pdf file of the manuscript.

---

## Round 0.2 · Minor Revisions

You have answered the comments well, and the paper is scientifically ready for publication. However, some small English edits remain (see my edited attached PDF) that need to be addressed before final acceptance.

---

## Round 0.3 · accepted · Accept

Thank you kindly for these final edits - the paper is now ready to be published. I look forward to seeing it online!